# Association of cesarean section and infectious outcomes among infants at 1 year of age: Logistic regression analysis using data of 104,065 records from the Japan Environment and Children's Study

**Hajime Maeda**[1]*, **Koichi Hashimoto**[1,2], **Hajime Iwasa**[3], **Hyo Kyozuka**[2,4], **Yohei Kume**[1], **Hayato Go**[1], **Akiko Sato**[2], **Yuka Ogata**[2], **Tsuyoshi Murata**[2,4], **Keiya Fujimori**[2,4], **Kosei Shinoki**[2], **Hidekazu Nishigori**[2,5], **Seiji Yasumura**[2,3], **Mitsuaki Hosoya**[1,2], **the Japan Environment and Children's Study (JECS) Group**[¶]

1 Department of Pediatrics, School of Medicine, Fukushima Medical University, Fukushima, Japan,
2 Fukushima Regional Center for the Japan Environmental and Children's Study, Fukushima, Japan,
3 Department of Public Health, School of Medicine, Fukushima Medical University, Fukushima, Japan,
4 Department of Obstetrics and Gynecology, School of Medicine, Fukushima Medical University School, Fukushima, Japan, 5 Fukushima Medical Center for Children and Women, Fukushima Medical University, Fukushima, Japan

¶ Membership of the Japan Environment and Children's Study Group is provided in the Acknowledgments.
* hmaeda@fmu.ac.jp

## Abstract

### Background

There has been a recent decrease in the prevalence of infectious diseases in children worldwide due to the usage of vaccines. However, the association between cesarean delivery and infectious diseases remains unclear. Here, we aimed to clarify the association between cesarean delivery and the development of infectious diseases.

### Methods

This study is a cross-sectional study. We used data from the Japan Environment and Children's Study, which is a prospective, nationwide, government-funded birth cohort study. The data of 104,065 records were included. Information about the mode of delivery, central nervous system infection (CNSI), otitis media (OM), upper respiratory tract infection (URTI), lower respiratory tract infection (LRTI), gastrointestinal infection (GI), and urinary tract infection (UTI) was obtained from questionnaires and medical records transcripts. Multiple logistic regression analysis was used to assess the association between cesarean delivery and CNSI, OM, URTI, LRTI, GI, and UTI risk.

### Results

We included a total of 74,477 subjects in this study, of which 18.4% underwent cesarean deliveries. After adjusting for the perinatal, socioeconomic, and postnatal confounding

**Data Availability Statement:** Data are unsuitable for public deposition because of ethical restrictions and the legal framework of Japan. It is prohibited by the Act on the Protection of Personal Information (Act No. 57 of May 30, 2003, amended on September 9, 2015) to publicly deposit data containing personal information. The Ethical Guidelines for Medical and Health Research Involving Human Subjects enforced by the Japan Ministry of Education, Culture, Sports, Science and Technology and the Ministry of Health, Labor, and Welfare also restrict the open sharing of epidemiological data. All inquiries about access to data should be sent to jecs-en@nies.go.jp. The person responsible for handling inquiries sent to this e-mail address is Dr. Shoji F. Nakayama, JECS Program Office, National Institute for Environmental Studies.

**Funding:** The authors received no specific funding for this work.

**Competing interests:** The authors have declared that no competing interests exist.

factors, children born by cesarean delivery did not have an increased risk of developing CNSI (95% confidence interval [CI] 0.46–1.35), OM (95% CI 0.99–1.12), URTI (95% CI 0.97–1.06), LRTI (95% CI 0.98–1.15), GI (95% CI 0.98–1.11), or UTI (95% CI 0.95–1.45).

## Conclusions

This nationwide cohort study did not find an association between cesarean delivery and CNSI, OM, URTI, LRTI, GI, and UTI. However, further studies are needed to evaluate the role of cesarean delivery in the development of infectious diseases.

## Introduction

The rates of infectious diseases in children have recently decreased worldwide, especially among children under the age of 5 years [1]. Globally, mortality among children under the age of 5 years has declined from 216.0 deaths per 1000 live births in 1950 to 38.9 deaths per 1000 live births in 2017, which was driven by global declines in deaths from diarrhea, lower respiratory tract infection (LRTI), and other common infectious diseases [2,3]. Nevertheless, there were still 5.4 million global deaths in children under the age of 5 years in 2017 [2]. In Japan, as the use of the *Haemophilus influenzae* type b (Hib) vaccine and pneumococcal conjugated vaccine (PCV) have become more widespread, the incidence of bacterial meningitis, pneumonia, and otitis media (OM) have decreased [4–7]. However, infectious diseases are still serious health concerns in Japan due to the significant impact on a child's health, hospitalization, and quality of life.

The rates of cesarean delivery have been increasing worldwide [8–10], and have approximately doubled in Japan since 1990, reaching 18.6% in 2014 [11]. Some recent reports described the association between cesarean delivery and infectious diseases [1,12,13]. Compared with children born by vaginal delivery, those born by cesarean delivery have different gut flora and cytokine profiles [14–18], which are thought to be related to infectious diseases; however, the mechanism is unclear. In contrast, some reports showed that cesarean delivery was not associated with the development of infectious diseases [12,19]. Therefore, the associations between cesarean delivery and infectious disease are controversial. In this study, we investigated the association between infants delivered by cesarean section and the development of infectious diseases using data from a large sample size cohort study of the Japan Environment and Children's Study (JECS).

## Materials and methods

### Study design

This study is a cross-sectional study. We used data from the JECS and investigated the association between cesarean delivery and the development of infectious diseases in infants. The detailed design of the JECS, a prospective nationwide government-funded birth cohort study, has been previously described [20]. The JECS investigates the effects of environmental factors on children's health by tracking mothers and their children until the children reach 13 years of age. From January 24, 2011 to March 31, 2014, 103,062 pregnancies were recruited to participate in the JECS at 15 Regional Centres in Japan (Hokkaido, Miyagi, Fukushima, Chiba, Kanagawa, Koshin, Toyama, Aichi, Kyoto, Osaka, Hyogo, Tottori, Kochi, Fukuoka, and South Kyushu/Okinawa). Each Regional Center consisted of one or more study areas to ensure

generalizability and the ability to extrapolate the results of the JECS to the Japanese population. Caregivers completed questionnaires regarding information about the mothers and their children during pregnancy and when the children were 6 and 12 months of age. The JECS protocol was reviewed and approved by the Ministry of the Environment's Institutional Review Board on Epidemiological Studies (No. 100910001) and Ethics Committees of all participating institutions. The JECS was conducted in accordance with the Declaration of Helsinki and other nationally valid regulations and guidelines. Written informed consent was obtained from all participants.

## Data collection

We used seven types of data from the dataset released in March 2018 (jecs-an-20180131) that included information from questionnaires and medical record transcripts. MT1 provided data from self-reported questionnaires collected during the first trimester that included information about the mothers' medical backgrounds. DrT1 provided medical information collected from the medical record transcripts during the first trimester provided by each co-operating health care provider. MT2 provided data from self-reported questionnaires collected during the second/third trimester that included information on lifestyle and socioeconomic status. Dr0m provided perinatal information collected from the medical record transcripts provided by each co-operating health care provider. M1m provided data from self-reported questionnaires collected at one month after birth. C6m provided data from self-reported questionnaires collected at six months after birth. C1Y provided data from self-reported questionnaires collected at 12 months after birth. We excluded women who miscarried, had multiple births, whose babies were stillborn, and whose data regarding confounders were missing (Fig 1).

## Outcomes, exposures, and covariates

Information about infectious diseases was obtained from the C1Y response to "Has your child ever been diagnosed by a doctor with the following infections?" The following diseases were targeted: central nervous system infection (CNSI), OM, upper respiratory tract infection (URTI), LRTI, gastrointestinal infection (GI), and urinary tract infection (UTI). CNSI was defined as the morbidity of encephalitis/encephalopathy, bacterial meningitis, or meningitis/aseptic meningitis. Infections were defined as the morbidity of CNSI, OM, URTI, GI, or UTI.

The mode of delivery was divided into cesarean sections and vaginal deliveries based on medical record transcripts in the Dr0m data. Participants were assigned to the cesarean or vaginal delivery group.

We considered the following factors as possible confounders in the regression analyses: perinatal factors including maternal age at pregnancy, parity, sex, preterm, small for gestational age (SGA), maternal allergy, maternal active or passive smoking during pregnancy, and maternal drinking during pregnancy: socioeconomic factors including maternal education periods, annual family income, and marital status: and postnatal factors including breastfeeding at six months, pet ownership, passive smoking exposure of infants after birth, children's allergy, sibling, nursery, and vaccination status. The maternal age at pregnancy (DrT1 data) was categorized as follows: < 20 years, 20–29 years, 30–39 years, and ≥ 40 years. Parity (DrT1 data) was categorized as 0, 1, or ≥ 2. Preterm was defined as <37 weeks' gestation. Standard deviation (SD) was calculated based on Japanese neonatal anthropometric charts [21], which accounted for GA, sex, and parity. Infants with values <-1.5 SDs were defined as SGA [22]. Maternal allergy history (MT1 data) was considered positive if a parent reported a history of asthma, allergic rhinitis, atopic dermatitis, allergic conjunctivitis, food allergies, drug allergies,

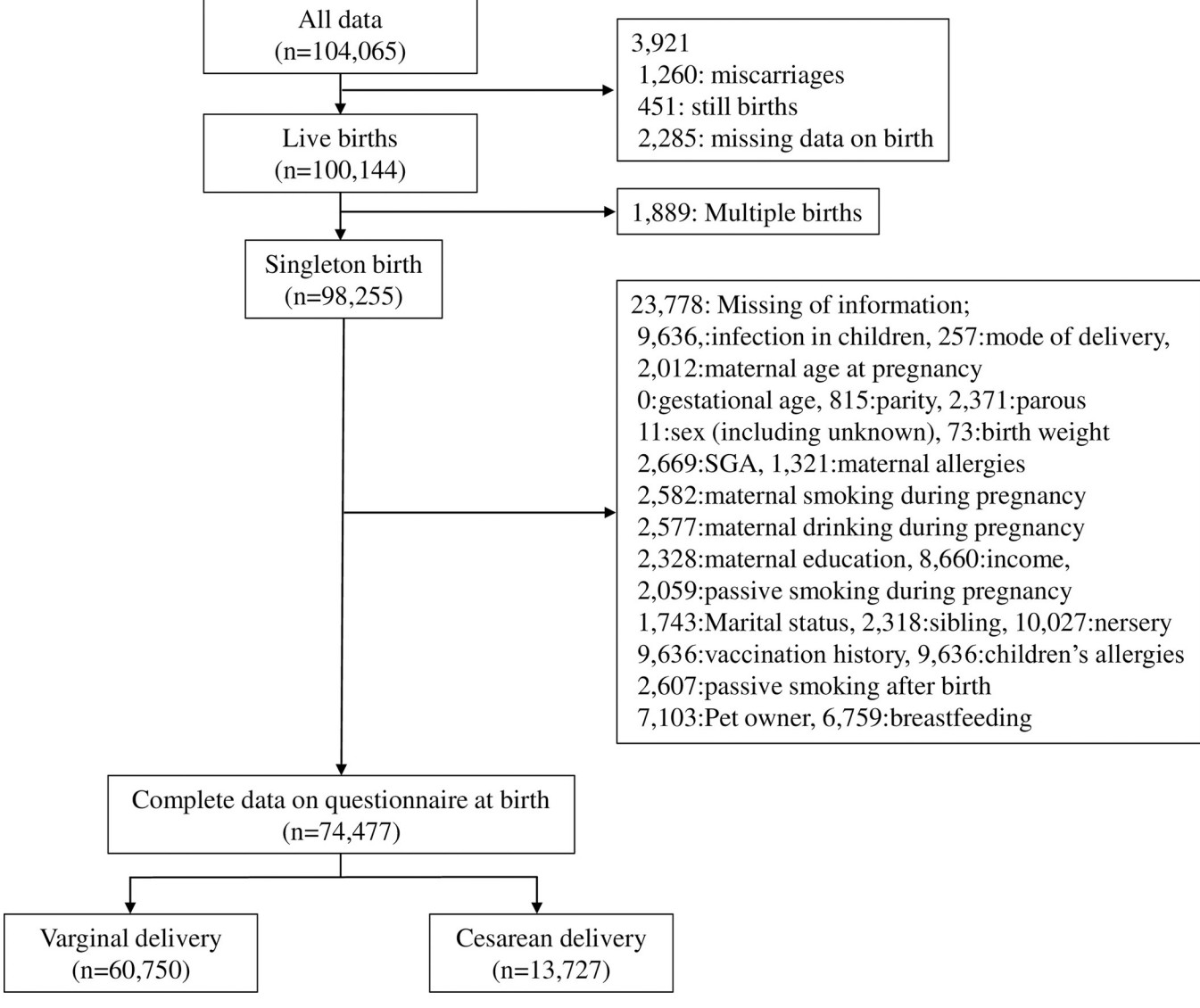

**Fig 1. Flow diagram of the sample population selected for analysis.**

contact dermatitis, or sick building syndrome. Data on the maternal smoking status and alcohol drinking habits were obtained from MT2.

The mothers' educational periods (MT2 data) were grouped as follows: junior high school, < 10 years; high school, 10–12 years; technical/vocational college or university, 13–16 years; and graduate school, ≥ 17 years. The participants' annual household incomes (MT2 data) were categorized as follows: < 2,000,000 Japanese yen (JPY), 2,000,000–5,999,999 JPY, 6,000,000–9,999,999 JPY, and ≥ 10,000,000 JPY. Data on marital status were obtained from MT1. Data on siblings and passive smoking exposure of infants were obtained from M1m. Data on breastfeeding and pet ownership were obtained from C6m. Data on children's allergy history, nursery, and vaccines were obtained from C1Y. Children's allergy history was considered positive if a parent reported a history of asthma, atopic dermatitis, food allergies, or infant's gastrointestinal allergies, allergic conjunctivitis, or allergic rhinitis.

## Statistical analyses

Characteristics of the mothers and their children were summarized according to the delivery mode. A chi-squared test was used to compare categorical variables. The effect size was evaluated using the phi coefficient. Multiple logistic regression was performed to determine the risk of CNSI, OM, URTI, LRTI, GI, and UTI associated with cesarean delivery by calculating the odds ratios (ORs), which were adjusted for the aforementioned confounders, and the 95% confidence intervals (CIs). We initially adjusted for perinatal and socioeconomic factors. We then adjusted for perinatal, socioeconomic, and postnatal factors. Statistical analyses were performed using Stata, version 15.0 (Stata Corporation LLC, College Station, TX, USA). $P$-values $< 0.05$ indicated statistical significance.

## Results

The jecs-an-20180131 dataset contained information about 104,065 fetuses and 98,255 live singleton births. After applying the study's inclusion criteria, 74,477 subjects were eligible to participate in this study (Fig 1). Of the 74,477 infants, 13,727 (18.4%) and 60,750 (81.6%) underwent cesarean and vaginal deliveries, respectively. Table 1 presents the participants' characteristics. A higher incidence of cesarean delivery was associated with a maternal age at pregnancy $\geq 30$ years, parity $\geq 2$, preterm birth, SGA, maternal active smoking during pregnancy, higher maternal education periods, higher annual family income, pet ownership, and rotavirus vaccine. A lower incidence of cesarean delivery was associated with primipara, breastfeeding at six months, children' allergy, and siblings.

Multiple logistic regression was performed to assess the risk of CNSI, OM, URTI, LRTI, GI, and UTI associated with cesarean delivery (Table 2). After adjusting for perinatal and socioeconomic confounding factors, cesarean delivery was not associated with an increased risk of CNSI (aOR = 0.78; 95% CI, 0.46–1.34), OM (aOR = 1.05; 95% CI, 0.99–1.11), URTI (aOR = 1.01; 95% CI, 0.97–1.06), LRTI (aOR = 1.07; 95% CI, 0.99–1.15), GI (aOR = 1.06; 95% CI, 0.99–1.13), or UTI (aOR = 1.19; 95% CI, 0.97–1.47). After adjusting for perinatal, socioeconomic, and postnatal confounding factors, cesarean delivery was not associated with an increased risk of CNSI (aOR = 0.79; 95% CI, 0.46–1.35), OM (aOR = 1.06; 95% CI, 0.99–1.12), URTI (aOR = 1.01; 95% CI, 0.97–1.06), LRTI (aOR = 1.06; 95% CI, 0.98–1.15), GI (aOR = 1.04; 95% CI, 0.98–1.11) or UTI (aOR = 1.17; 95% CI, 0.95–1.45). After adjusting for perinatal and socioeconomic confounding factors, cesarean delivery was significantly associated with an increased risk of infection (aOR = 1.04; 95% CI, 1.00–1.09). After adjusting for perinatal, socioeconomic, and postnatal confounding factors, cesarean delivery was significantly associated with an increased risk of infection (aOR = 1.05; 95% CI, 1.00–1.09).

## Discussion

In this study, no associations were found between cesarean delivery and the development of CNSI, OM, URTI, LRTI, GI, and UTI among infants aged one year. Cesarean delivery was significantly associated with an increased risk of infection. However, the phi coefficient between the two groups was 0.0089 and the difference was interpreted as very small. Such a small difference was considered significant because of the large sample size and detection power in this study. Moreover, the ideal sample size has an event-predictor ratio of 10:1 or greater. Therefore, this was not a problem in this study because of its large sample size. Several studies have investigated the association between cesarean delivery and infectious diseases. A cohort study in Norway in 2011 showed no increased risk of recurrent LRTIs in children under the age of 36 months delivered by cesarean section (relative risk = 1.07; 95% CI, 0.92–1.25) [19]. A retrospective population-based cohort study in Australia in 2011 showed children delivered by

**Table 1. Characteristics of the study participants.**

| | Vaginal delivery (n = 60,750) n | % | Cesarean delivery (n = 13,727) n | % | p value |
|---|---|---|---|---|---|
| Maternal age at pregnancy (years) | | | | | <0.001 |
| <20 | 411 | 0.7 | 50 | 0.4 | |
| 20–29 | 24,380 | 40.1 | 3,895 | 28.4 | |
| 30–39 | 34,177 | 56.3 | 8,879 | 64.7 | |
| >= 40 | 1,782 | 2.9 | 903 | 6.6 | |
| Parity | | | | | 0.002 |
| 0 | 18,148 | 29.9 | 3,911 | 28.5 | |
| 1 | 21,084 | 34.7 | 4,769 | 34.7 | |
| >= 2 | 21,518 | 35.4 | 5,047 | 36.8 | |
| Sex (male) | 31,061 | 51.1 | 7,037 | 51.3 | 0.776 |
| Preterm | 1,828 | 3.0 | 1,373 | 10.0 | <0.001 |
| SGA | 1,880 | 3.1 | 671 | 4.9 | <0.001 |
| Maternal allergy | 31,311 | 51.5 | 7,094 | 51.7 | 0.769 |
| Maternal active smoking during pregnancy | 2,216 | 3.6 | 570 | 4.2 | 0.005 |
| Maternal passive smoking during pregnancy | 21,601 | 35.6 | 4,998 | 36.4 | 0.06 |
| Maternal drinking during pregnancy | 1,709 | 2.8 | 348 | 2.5 | 0.074 |
| Maternal education periods (year) | | | | | 0.009 |
| <10 | 2,280 | 3.8 | 519 | 3.8 | |
| 10–12 | 18,911 | 31.1 | 4,438 | 32.3 | |
| 13–16 | 38,623 | 63.6 | 8,531 | 62.1 | |
| >= 17 | 936 | 1.5 | 239 | 1.7 | |
| Annual family income (JPY) | | | | | 0.009 |
| <2,000,000 | 3,048 | 5.0 | 740 | 5.4 | |
| 2,000,000–5,999,999 | 41,204 | 67.8 | 9,122 | 66.5 | |
| 6,000,000–9,999,999 | 13,911 | 22.9 | 3,230 | 23.5 | |
| >= 10,000,000 | 2,587 | 4.3 | 635 | 4.6 | |
| Marital status | 58,654 | 96.5 | 13,272 | 96.7 | 0.43 |
| Breastfeeding at 6 months | 46,216 | 76.1 | 9,743 | 71.0 | <0.001 |
| Pet ownership | 14,306 | 23.5 | 3,371 | 24.6 | 0.012 |
| Passive smoking exposure of infants after birth | 30,672 | 50.5 | 6,885 | 50.2 | 0.482 |
| Children's allergy | 7,552 | 12.4 | 1,609 | 11.7 | 0.022 |
| Sibling | 33,937 | 55.9 | 7,337 | 53.5 | <0.001 |
| Nursery | 16,455 | 27.1 | 3,759 | 27.4 | 0.479 |
| Vaccines | | | | | |
| DPT vaccine | 54,458 | 89.6 | 12,323 | 89.8 | 0.653 |
| Hib vaccine | 57,842 | 95.2 | 13,073 | 95.2 | 0.911 |
| Pneumococcal vaccine | 56,752 | 93.4 | 12,845 | 93.6 | 0.505 |
| Rotavirus vaccine | 26,364 | 43.4 | 6,128 | 44.6 | 0.008 |
| influenza vaccine | 10,890 | 17.9 | 2,552 | 18.6 | 0.067 |
| Infection | 22,729 | 37.4 | 5,287 | 38.5 | 0.015 |
| CNSI | 94 | 0.2 | 16 | 0.1 | 0.293 |
| OM | 7,064 | 11.6 | 1,642 | 12.0 | 0.272 |
| URTI | 14,161 | 23.3 | 3,283 | 23.9 | 0.13 |
| LRTI | 3,751 | 6.2 | 888 | 6.5 | 0.197 |
| GI | 5,597 | 9.2 | 1,323 | 9.6 | 0.122 |

(*Continued*)

**Table 1.** (Continued)

| | Vaginal delivery (n = 60,750) n | % | Cesarean delivery (n = 13,727) n | % | *p* value |
|---|---|---|---|---|---|
| UTI | 434 | 0.7 | 119 | 0.9 | 0.06 |

SGA: small gestational age; JPY: Japanese yen; DPT: diphtheria, pertussis, and tetanus; Hib: haemophilus influenza type b; CNSI: central nervous system infection; OM: otitis media; URTI: Upper respiratory tract infection; LRTI: Lower respiratory tract infection; GI: gastrointestinal infection; UTI: urinary tract infection.

The P value was determined using the chi square test.

elective cesarean section had an increased risk of hospitalizations for bronchiolitis at age < 12 months (incidence rate ratio (IRR) = 1.11; 95% CI, 1.01–1.23) and 12–23 months (IRR = 1.20; 95% CI, 0.94–1.53) but no association with the risk of hospitalizations for pneumonia at age < 12 months (IRR = 1.03; 95% CI, 0.80–1.33) and 12–23 months (IRR = 1.09; 95% CI, 0.88–1.34) [12]. A prospective birth cohort study in Denmark in 2018 showed an association of LRTI with cesarean section (adjusted IRR = 1.49; 95% CI, 1.12–1.99) [1]. A population-based cohort study in Denmark in 2007 showed an independent effect of cesarean section (IRR = 1.29; 95% CI, 1.12–1.49) on the incidence rate of viral meningitis [13]. These results are controversial. However, previous studies examining the association between cesarean delivery and infectious diseases varied in sample size, age, follow-up period, case definition, and adjustment of confounders. This was a large nationwide cohort study in which the association between cesarean delivery and the development of CNSI, OM, URTI, LRTI, GI, and UTI was assessed while adjusting for potential confounders.

The mechanism underlying the association between cesarean delivery and infectious diseases is unclear. Children born by vaginal delivery are exposed to diverse microbiological flora from the birth canal. Children born by cesarean delivery show delays and differences in the establishment of their gut flora and altered cytokine profiles [14]. Although infants delivered vaginally harbor bacterial communities resembling those of their mother's vaginas, infants born by cesarean delivery are enriched with skin microbiota [15,16]. Moreover, subsequent intestinal and airway colonization alters immunomodulation and susceptibility to LRTI [17,18]. These mechanisms may affect elective cesarean deliveries more than emergency cesarean deliveries because emergency cesarean deliveries often occur after the onset of labor,

**Table 2. Risks of infection diseases associated with cesarean delivery.**

| | | Infection | CNSI | OM | URTI | LRTI | GI | UTI |
|---|---|---|---|---|---|---|---|---|
| Vaginal delivery | | Ref | Ref | Ref | Ref | Ref | Ref | Ref |
| Cesarean delivery | cOR 95% CI | 1.05 (1.01–1.09) | 0.75 (0.44–1.28) | 1.03 (0.98–1.09) | 1.03 (0.99–1.08) | 1.05 (0.97–1.13) | 1.05 (0.99–1.12) | 1.22 (0.99–1.49) |
| | aOR*1 95% CI | 1.04 (1.00–1.09) | 0.78 (0.46–1.34) | 1.05 (0.99–1.11) | 1.01 (0.97–1.06) | 1.07 (0.99–1.15) | 1.06 (0.99–1.13) | 1.19 (0.97–1.47) |
| | aOR*2 95% CI | 1.05 (1.00–1.09) | 0.79 (0.46–1.35) | 1.06 (0.99–1.12) | 1.01 (0.97–1.06) | 1.06 (0.98–1.15) | 1.04 (0.98–1.11) | 1.17 (0.95–1.45) |

Ref: reference; cOR: crude odds ratio; aOR: adjusted odds ratio; CI: confidence interval; CNSI: central nervous system infection. OM: otitis media; URTI: Upper respiratory tract infection; LRTI: Lower respiratory tract infection; UTI: urinary tract infection; GI: gastrointestinal infection.

*1a was adjusted with potential confounders, including maternal age at pregnancy, parity, sex, preterm, small for gestational age, maternal allergy history, maternal active and passive smoking during pregnancy, maternal drinking during pregnancy, maternal educational periods, annual family income, and marital status.

*2; The model was adjusted with breastfeeding at 6 months, pet ownership, passive smoking exposure of infants after birth, children's allergies, siblings, nursery, and vaccine status in addition to the above confounders of aOR*1.

potentially resulting in exposure to vaginal microflora and both maternal and fetal stress [23]. Hence, the delivery mode may be a crucial factor influencing the incidence of disease. However, the results of this study do not support this hypothesis. Based on recent evidence of the presence of bacteria in the placenta, amniotic fluid, and meconium, some investigators believe that the microbiome may be seeded before birth [24]. These findings may support the results of this study showing no association between cesarean delivery and the development of infectious diseases.

The strengths of our study include the prospective and nationwide cohort design; the comparison of maternal questionnaires with medical records to verify the exposure variables and other covariates; and use of prospectively collected data regarding CNSI, OM, URTI, LRTI, GI, and UTI. Multiple analyses were performed adjusting for perinatal, socioeconomic, and postnatal factors.

However, this study has several limitations. First, we evaluated infectious diseases at one year of age using participants' self-reported questionnaires, which may have led to the underreporting of infectious diseases. Second, vaccination rates were high in this study, so different results may be obtained in areas with low vaccination. Third, we did not distinguish between emergency and elective cesarean sections. Finally, there was no information on the severity of infectious diseases, so we were unable to evaluate the association between cesarean section and the severity of infectious diseases. Despite these limitations, our study evaluated data from a large, nationwide prospective birth cohort study that has maintained high follow-up and questionnaire response rates [20]; therefore, our study provides strong evidence against an association between cesarean delivery and infectious diseases. This may have important clinical and public health implications. If cesarean delivery has clinical benefits, it should not be avoided because of the risk of infection in the infant.

## Conclusions

The findings of this study, based on a large nationwide cohort, revealed no association between cesarean delivery and the development of neonatal infectious diseases. Further studies are needed to evaluate the role of cesarean delivery in the development of infectious diseases in infants.

## Acknowledgments

The findings and conclusions of this article are solely the responsibility of the authors and do not represent the official views of the Ministry of the Environment, Japan. We thank all the participants and staff involved in the Japan Environment and Children's Study. Members of the JECS Group as of 2020: Michihiro Kamijima (principal investigator, Nagoya City University, Nagoya, Japan), Shin Yamazaki (National Institute for Environmental Studies, Tsukuba, Japan), Yukihiro Ohya (National Center for Child Health and Development, Tokyo, Japan), Reiko Kishi (Hokkaido University, Sapporo, Japan), Nobuo Yaegashi (Tohoku University, Sendai, Japan), Koichi Hashimoto (Fukushima Medical University, Fukushima, Japan), Chisato Mori (Chiba University, Chiba, Japan), Shuichi Ito (Yokohama City University, Yokohama, Japan), Zentaro Yamagata (University of Yamanashi, Chuo, Japan), Hidekuni Inadera (University of Toyama, Toyama, Japan), Takeo Nakayama (Kyoto University, Kyoto, Japan), Hiroyasu Iso (Osaka University, Suita, Japan), Masayuki Shima (Hyogo College of Medicine, Nishinomiya, Japan), Youichi Kurozawa (Tottori University, Yonago, Japan), Narufumi Suganuma (Kochi University, Nankoku, Japan), Koichi Kusuhara (University of Occupational and Environmental Health, Kitakyushu, Japan), and Takahiko Katoh (Kumamoto University, Kumamoto, Japan).

## Author Contributions

**Conceptualization:** Hajime Maeda.

**Data curation:** Akiko Sato, Yuka Ogata.

**Formal analysis:** Hajime Maeda, Hajime Iwasa.

**Investigation:** Akiko Sato, Yuka Ogata.

**Project administration:** Koichi Hashimoto, Hyo Kyozuka, Akiko Sato, Yuka Ogata, Kosei Shinoki.

**Supervision:** Koichi Hashimoto, Hajime Iwasa.

**Writing – original draft:** Hajime Maeda.

**Writing – review & editing:** Koichi Hashimoto, Hajime Iwasa, Hyo Kyozuka, Yohei Kume, Hayato Go, Tsuyoshi Murata, Keiya Fujimori, Kosei Shinoki, Hidekazu Nishigori, Seiji Yasumura, Mitsuaki Hosoya.

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
