## [Decision Letter · Decision Letter 0]

10 Nov 2023

PONE-D-23-20377Association of cesarean section and infectious outcomes among infants at 1 year of age: Logistic regression analysis using data of 104,065 records from the Japan Environment and Children’s StudyPLOS ONE

Dear Dr. Maeda,

Thank you for submitting your manuscript to PLOS ONE. After careful consideration, we feel that it has merit but does not fully meet PLOS ONE’s publication criteria as it currently stands. Therefore, we invite you to submit a revised version of the manuscript that addresses the points raised during the review process.

We look forward to receiving your revised manuscript.

Kind regards,

Kazumichi Fujioka

Academic Editor

PLOS ONE

Journal Requirements:

- http://dx.doi.org/10.1136/bmjopen-2017-017086

In your revision ensure you cite all your sources (including your own works), and quote or rephrase any duplicated text outside the methods section. Further consideration is dependent on these concerns being addressed.

Additional Editor Comments :

At first, I am really sorry for this delay of editorial response.

To be honest, we have been declined by more than 40 potential reviewers until now....

Please revise according to the reviewers suggestion very carefully. It is hard to find further reviewer if the reviewers are not satisfied with your revision...

Reviewers' comments:

Reviewer's Responses to Questions

**Comments to the Author**

1. Is the manuscript technically sound, and do the data support the conclusions?

Reviewer #1: Yes

Reviewer #2: Yes

2. Has the statistical analysis been performed appropriately and rigorously? 

Reviewer #1: Yes

Reviewer #2: Yes

3. Have the authors made all data underlying the findings in their manuscript fully available?

Reviewer #1: Yes

Reviewer #2: Yes

4. Is the manuscript presented in an intelligible fashion and written in standard English?

Reviewer #1: Yes

Reviewer #2: Yes

5. Review Comments to the Author

Reviewer #1: In this manuscript, the authors investigated the association between caesarean section and infectious diseases in children in Japan using the Japan Environment and Children's Study (JECS). The manuscript is well written and well organised. This study is important and interesting because there is controversy about this association. However, I found some problems as listed below, which, if addressed, will improve the manuscript.

Major

1) Study design, measurement and reporting of association

On page 28, line 253, the authors describe this study as a "prospective and nationwide cohort design". The study design should be clearly stated in the title, abstract and material and methods. If this is a prospective cohort study, why did the authors report the measure of association using odds ratio rather than relative risk?

2) Outcome description

It was often difficult to understand what population the authors wanted to know about infections. Of course, I could imagine from the title and context that the authors wanted to focus on infectious diseases in children from birth to one year, but this information should be clearly explained. On page 29, line 273-274, the authors concluded that "further studies are needed to evaluate the role of caesarean delivery in the development of neonatal infectious diseases", and this caused me further confusion because this article was not only about neonatal infectious diseases.

3) Validity of key information

Although the authors explain in the limitation about the potential underestimation of infectious disease, but this should be discussed further. From the "Data collection" section, all information after the neonatal period was from self-reported questionnaires. The information on infectious diseases is essential in this study. The authors should explain the details of the validity of the information collected by questionnaires and how the result could be influenced There was 'under-reporting' as described by the authors.

(It may be useful to cite an article explaining the validation of the JECS, if available)

Minor

1) Page 13, line 113-114

The word "morbidity" should be replaced by another word if the authors meant simple incidence of infectious disease rather than death.

2) Page 17, lines 176-179.

It is appropriate to describe the phi coefficient here in the results section, but its assessment should be discussed in the discussion section.

Reviewer #2: This report uses data from the Japan Environment and Children's Study to show that there is no association between cesarean section and infant infections. This is a valuable report that shows the situation in Japan based on large amounts of data, but some corrections are needed.

Major comments

1. Regarding infections up to 1 year of age, are confounding factors whether the child has an underlying disease or whether the cesarean section was elective or emergency? Other papers report investigations limited to the elective cesarean section. I think the author should discuss the reason for not including it as an adjustment factor or list it in the limitations section.

2. L206-238 How are these sentences related to the results of the authors’ study? I believe it is simply explaining the efficacy of vaccines and the epidemiology of each infectious disease.

Minor comments

1. L79-81 Why do authors also classify by region (e.g., Koshin and South Kyusyu/Okinawa) instead of the prefecture name? Is it differentiated based on the distribution of the number of cases? Please explain.

2. L109-112 Is there a definition for each infectious disease?

Furthermore, the reviewer could not understand the meaning of the statement “based on the doctor’s self-reported diagnoses and information given participants’ questionnaires.” Does the database also include the data from a questionnaire survey among doctors regarding their diagnosis? Or did the family select each disease on a questionnaire based on the doctor's diagnosis?

3. L117-122 As in the table, please also indicate whether each question is perinatal socioeconomic or postnatal confounding in the Method section.

4. L175-176 Why not did the authors mention the results of aOR*1 regarding the relationship between overall infections and cesarean sections, as well as the results of aOR*2? Please let me know if there is a reason.

5. L188-200 This section introduces articles from other countries that reported a relationship between cesarean sections and infectious diseases, unlike the authors’ study, and describes "These results are inconsistent with our results." Thus, reference 19 reported that there is no relationship with infectious diseases, similar to the author's research, so I think the sentences (L188-191) are inappropriate for this part of the article.

6. L268 Based on the results of this research, I think "infant" is correct rather than "neonatal."

6. PLOS authors have the option to publish the peer review history of their article (what does this mean?). If published, this will include your full peer review and any attached files.

Reviewer #1: **Yes: **Shota Myojin

Reviewer #2: No

---

## [Author Response · Author response to Decision Letter 0]

11 Dec 2023

Responses to the Reviewers

Manuscript ID: PONE-D-23-20377

We wish to re-submit the manuscript titled “Association of cesarean section and infectious outcomes among infants at 1 year of age: Logistic regression analysis using data of 104,065 records from the Japan Environment and Children’s Study.” 

We thank the reviewers for their insightful review and valuable comments, which have helped us improve our manuscript significantly. We have addressed the major concerns of the reviewers and provided a point-by-point response to the reviewers’ comments below. 

Reviewer #1: In this manuscript, the authors investigated the association between caesarean section and infectious diseases in children in Japan using the Japan Environment and Children's Study (JECS). The manuscript is well written and well organized. This study is important and interesting because there is controversy about this association. However, I found some problems as listed below, which, if addressed, will improve the manuscript.

Major

1) Study design, measurement and reporting of association

On page 28, line 253, the authors describe this study as a "prospective and nationwide cohort design". The study design should be clearly stated in the title, abstract and material and methods. If this is a prospective cohort study, why did the authors report the measure of association using odds ratio rather than relative risk?

Author response: Thank you for your valuable comment. Prospective cohort studies have the advantage of using a variety of measures. However, relative risk cannot be used in case-control studies. Therefore, the odds ratio is used as an approximation of the relative risk in case-control studies. In this cohort prospective study, we investigated the independent association between cesarean sections and infectious diseases. Therefore, we performed a logistic regression analysis to control for confounding factors and calculated the odds ratios. We have revised the text of the manuscript as follows:

Page 3, lines 30–31

This study used data from the Japan Environment and Children’s Study which is a prospective, nationwide, government-funded birth cohort study.

Page 5, lines 75–76

We used data from the JECS, a prospective nationwide government-funded birth cohort study [20].

2) Outcome description

It was often difficult to understand what population the authors wanted to know about infections. Of course, I could imagine from the title and context that the authors wanted to focus on infectious diseases in children from birth to one year, but this information should be clearly explained. On page 29, line 273-274, the authors concluded that "further studies are needed to evaluate the role of caesarean delivery in the development of neonatal infectious diseases", and this caused me further confusion because this article was not only about neonatal infectious diseases.

Author response: We apologize for this confusion. We focused on infectious diseases in infants. We have revised the text of the manuscript as follows:

Page 22, lines 250–252

Further studies are needed to evaluate the role of cesarean delivery in the development of infectious diseases in infants. 

3) Validity of key information

Although the authors explain in the limitation about the potential underestimation of infectious disease, but this should be discussed further. From the "Data collection" section, all information after the neonatal period was from self-reported questionnaires. The information on infectious diseases is essential in this study. The authors should explain the details of the validity of the information collected by questionnaires and how the result could be influenced There was 'under-reporting' as described by the authors.

(It may be useful to cite an article explaining the validation of the JECS, if available)

Author response: Thank you for your valuable comments.

In the JECS plan, the follow-up rate was to be 80% or higher at the end of the follow-up period. It is highly commendable that the follow-up rate has remained high (93.6% on average nationwide) as of September 2022. Moreover, one of the most important issues in making the results of the JECS survey more reliable is maintaining a high participant questionnaire recovery rate. Eighty percent retention is the target for the JECS. The average recovery rate for 1-year-olds was 91.4%. We have revised the text of the manuscript as follows:

Page 21-22, lines 240–244

Despite these limitations, our study evaluated data from a large, nationwide prospective birth cohort study that has maintained high follow-up and questionnaire response rates [20]; therefore, our study provides strong evidence against an association between cesarean delivery and infectious diseases.

Minor

1) Page 13, line 113-114

The word "morbidity" should be replaced by another word if the authors meant simple incidence of infectious disease rather than death. 

Author response: Thank you for your valuable comment. We used “morbidity” to mean the incidence of infectious diseases. We would use “mortality” if we wanted to refer to death.

2) Page 17, lines 176-179.

It is appropriate to describe the phi coefficient here in the results section, but its assessment should be discussed in the discussion section.

Author response: Thank you for your valuable comment. We have revised the text of the manuscript as follows:

Page 19, lines 189–193

Cesarean delivery was significantly associated with an increased risk of infection. However, the phi coefficient between the two groups was 0.0089, and the difference was interpreted as very small. Such a small difference was considered significant because of the large sample size and detection power of this study.

Reviewer #2: This report uses data from the Japan Environment and Children's Study to show that there is no association between cesarean section and infant infections. This is a valuable report that shows the situation in Japan based on large amounts of data, but some corrections are needed.

Major comments

1. Regarding infections up to 1 year of age, are confounding factors whether the child has an underlying disease or whether the cesarean section was elective or emergency? Other papers report investigations limited to the elective cesarean section. I think the author should discuss the reason for not including it as an adjustment factor or list it in the limitations section.

Author response: Thank you for your valuable comment. We have added the following text and reference to the revised manuscript:

Page 20, lines 219–222

These mechanisms may affect elective cesarean deliveries more than emergency cesarean deliveries because emergency cesarean deliveries often occur after the onset of labor, potentially resulting in exposure to vaginal microflora and maternal and fetal stress [23].

Page 21, lines 237–238

Third, we did not distinguish between emergency and elective cesarean sections.

References, Page 26, lines 340-342

23. Rusconi F, Zugna D, Annesi-Maesano I, Baïz N, Barros H, Correia S, et al. Mode of Delivery and Asthma at School Age in 9 European Birth Cohorts. Am J Epidemiol. 2017; 185: 465-473.

2. L206-238 How are these sentences related to the results of the authors’ study? I believe it is simply explaining the efficacy of vaccines and the epidemiology of each infectious disease.

Author response: Thank you for your valuable comment. We have removed lines 206–238 and references 23–34 from the text.

Minor comments

1. L79-81 Why do authors also classify by region (e.g., Koshin and South Kyusyu/Okinawa) instead of the prefecture name? Is it differentiated based on the distribution of the number of cases? Please explain.

Author response: Thank you for your valuable comment. To ensure generalizability and the ability to extrapolate the JECS results to the Japanese population, 15 regional centers covering a wide geographical area were selected. The urbanization and land development of the study locations are diverse. The urbanization status ranged from urban and suburban to rural, and the land development purposes ranged from agricultural and fishery to commercial and industrial uses. Regional centers were selected through a competitive process in which universities and other research institutions were invited to submit proposals for covered areas and populations, recruitment methods, organizational structures, regional liaison, and resources. Each regional center consists of one or more study areas. The population of the selected study areas ranged from 130,000 to 600,000. Assuming a birth rate of 1% in study areas, each regional center will have 1,300 to 6,000 annual births, or 4,400 annual births on average. The JECS aims to cover half of the births in the area. The selected regional centers are required to recruit 3,000 to 9,000 pregnant women in three years, totaling 100,000 participants from the 15 regional centers. We have added the following text to the revised manuscript:

Page 5, lines 81–83

Each regional center consisted of one or more study areas to ensure generalizability and the ability to extrapolate the results of the JECS to the Japanese population.

2. L109-112 Is there a definition for each infectious disease?

Furthermore, the reviewer could not understand the meaning of the statement “based on the doctor’s self-reported diagnoses and information given participants’ questionnaires.” Does the database also include the data from a questionnaire survey among doctors regarding their diagnosis? Or did the family select each disease on a questionnaire based on the doctor's diagnosis?

Author response: Thank you for your valuable comment. The family selected each disease on the questionnaire based on the doctor’s diagnosis. We have revised the text of the manuscript as follows:

Page 7, 110–113

Central nervous system infection (CNSI), OM, upper respiratory tract infection (URTI), LRTI, gastrointestinal infection (GI), and urinary tract infection (UTI) were assessed based on the information given in the participants’ questionnaires when the children were one year old (C1Y data).

3. L117-122 As in the table, please also indicate whether each question is perinatal socioeconomic or postnatal confounding in the Method section.

Author response: Thank you for your valuable comment. We have revised the text of the manuscript as follows:

Page 8, lines 117–124

We considered the following factors as possible confounders in the regression analyses: perinatal factors including maternal age at pregnancy, parity, sex, preterm, small for gestational age (SGA), maternal allergy, maternal active or passive smoking during pregnancy, and maternal drinking during pregnancy; socioeconomic factors including maternal education periods, annual family income, and marital status; and postnatal factors including breastfeeding at six months, pet ownership, passive smoking exposure of infants after birth, children’s allergy, sibling, nursery, and vaccination status.

4. L175-176 Why not did the authors mention the results of aOR*1 regarding the relationship between overall infections and cesarean sections, as well as the results of aOR*2? Please let me know if there is a reason.

Author response: Thank you for your valuable comment. We have revised the text of the manuscript as follows:

Page 11, lines 177–181

After adjusting for perinatal and socioeconomic confounding factors, cesarean delivery was significantly associated with an increased risk of infection (aOR = 1.04; 95% CI, 1.00–1.09). After adjusting for perinatal, socioeconomic, and postnatal confounding factors, cesarean delivery was significantly associated with an increased risk of infection (aOR = 1.05; 95% CI, 1.00–1.09).

5. L188-200 This section introduces articles from other countries that reported a relationship between cesarean sections and infectious diseases, unlike the authors’ study, and describes "These results are inconsistent with our results." Thus, reference 19 reported that there is no relationship with infectious diseases, similar to the author's research, so I think the sentences (L188-191) are inappropriate for this part of the article.

Author response: Thank you for your valuable comment. We have revised the text of the manuscript as follows:

Page 19, line 206

These results are controversial.

6. L268 Based on the results of this research, I think "infant" is correct rather than "neonatal."

Author response: Thank you for your valuable comment. We have revised the text of the manuscript as follows:

Page 22, lines 244–246

If cesarean delivery has clinical benefits, it should not be avoided because of the risk of infection in the infant.

Journal Requirements:

Author response: We have ensured that our manuscript meets PLOS ONE’s style requirements.

Author response: The data are unsuitable for public deposition due to ethical restrictions and Japan’s legal framework. The Act on the Protection of Personal Information (Act No. 57 of May 30, 2003; amended on September 9, 2015) prohibits the public deposition of data containing personal information. The Ethical Guidelines for Medical and Health Research Involving Human Subjects enforced by the Japan Ministry of Education, Culture, Sports, Science, and Technology and the Ministry of Health, Labor, and Welfare also restrict the open sharing of epidemiological data. All inquiries regarding data access should be sent to jecs-en@nies.go.jp. Dr. Shoji F. Nakayama of the JECS Program Office, National Institute for Environmental Studies, is responsible for handling the inquiries sent to this e-mail address.

- http://dx.doi.org/10.1136/bmjopen-2017-017086

In your revision ensure you cite all your sources (including your own works), and quote or rephrase any duplicated text outside the methods section. Further consideration is dependent on these concerns being addressed.

Author response: In this publication, “Associations of caesarean delivery and the occurrence of neurodevelopmental disorders, asthma or obesity in childhood based on Taiwan birth cohort study” the authors investigated the association between cesarean sections and neurodevelopmental disorders, asthma, or obesity in children in Taiwan using the Taiwan Birth Cohort Study (TBCS). We investigated the association between cesarean sections and infectious diseases in children in Japan using the Japan Environment and Children's Study (JECS). The outcomes differed between our study and this previous study. Therefore, we did not cite this publication. The entire text has been carefully read. However, we could not find any text overlapping with this publication.

Author response: The data are unsuitable for public deposition due to ethical restrictions and Japan’s legal framework. The Act on the Protection of Personal Information (Act No. 57 of May 30, 2003; amended on September 9, 2015) prohibits the public deposition of data containing personal information. The Ethical Guidelines for Medical and Health Research Involving Human Subjects enforced by the Japan Ministry of Education, Culture, Sports, Science, and Technology and the Ministry of Health, Labor, and Welfare also restrict the open sharing of epidemiological data. All inquiries regarding data access should be sent to jecs-en@nies.go.jp. Dr. Shoji F. Nakayama of the JECS Program Office, National Institute for Environmental Studies, is responsible for handling the inquiries sent to this e-mail address.

---

## [Decision Letter · Decision Letter 1]

26 Dec 2023

PONE-D-23-20377R1Association of cesarean section and infectious outcomes among infants at 1 year of age: Logistic regression analysis using data of 104,065 records from the Japan Environment and Children’s StudyPLOS ONE

Dear Dr. Maeda,

Thank you for submitting your manuscript to PLOS ONE. After careful consideration, we feel that it has merit but does not fully meet PLOS ONE’s publication criteria as it currently stands. Therefore, we invite you to submit a revised version of the manuscript that addresses the points raised during the review process.

**ACADEMIC EDITOR: **

**Rev 2 is OK for acceptance, but Rev 1 is still concerned about the revised manuscript.**

**Please correct precisely following the Reviewers comment.**

We look forward to receiving your revised manuscript.

Kind regards,

Kazumichi Fujioka

Academic Editor

PLOS ONE

Journal Requirements:

Reviewers' comments:

Reviewer's Responses to Questions

**Comments to the Author**

1. If the authors have adequately addressed your comments raised in a previous round of review and you feel that this manuscript is now acceptable for publication, you may indicate that here to bypass the “Comments to the Author” section, enter your conflict of interest statement in the “Confidential to Editor” section, and submit your "Accept" recommendation.

Reviewer #1: (No Response)

Reviewer #2: All comments have been addressed

2. Is the manuscript technically sound, and do the data support the conclusions?

Reviewer #1: Partly

Reviewer #2: Yes

3. Has the statistical analysis been performed appropriately and rigorously? 

Reviewer #1: N/A

Reviewer #2: Yes

4. Have the authors made all data underlying the findings in their manuscript fully available?

Reviewer #1: Yes

Reviewer #2: Yes

5. Is the manuscript presented in an intelligible fashion and written in standard English?

Reviewer #1: Yes

Reviewer #2: Yes

6. Review Comments to the Author

Reviewer #1: Although most of the comments were reviewed and corrected appropriately, this article still has ambiguous parts.

Major:

1. Study design

For my comment about the study design in the first revision, the authors answered that this study is a "case-control study". However, the authors did not mention this anywhere in the manuscript and this is confusing for readers. From my understanding, this is a nested case-control study using an existing cohort. The section on "Study design" does not explain the study design of this study; this is only an explanation of the JECS. The study design of 'this study' should be clarified. Additionally, please clearly describe the definition of case and control.

2. Sample size calculation

How did the authors estimate the sample size? Did the authors try to do matching cases and controls to increase the power?

In a nested case-control study, controls are selected by incidence density sampling or sampling from baseline subcohort. Either way, random sampling for controls is used.

It seems that the authors used all cases and all controls available in the main analysis, but how will the result be affected if they used adequate sample size calculation?

Reviewer #2: Thank you for your revision. Following the comments from Reviewer 2, the authors have appropriately revised the manuscript.

7. PLOS authors have the option to publish the peer review history of their article (what does this mean?). If published, this will include your full peer review and any attached files.

Reviewer #1: No

Reviewer #2: No

---

## [Author Response · Author response to Decision Letter 1]

12 Jan 2024

Responses to the Reviewers

Manuscript ID: PONE-D-23-20377

We wish to re-submit the manuscript titled “Association of cesarean section and infectious outcomes among infants at 1 year of age: Logistic regression analysis using data of 104,065 records from the Japan Environment and Children’s Study.” 

We thank the reviewers for their insightful review and valuable comments, which have helped us improve our manuscript significantly. We have addressed the major concerns of the reviewers and provided a point-by-point response to the reviewers’ comments below. 

Reviewer #1: Although most of the comments were reviewed and corrected appropriately, this article still has ambiguous parts.

Major

1. Study design

For my comment about the study design in the first revision, the authors answered that this study is a "case-control study". However, the authors did not mention this anywhere in the manuscript and this is confusing for readers. From my understanding, this is a nested case-control study using an existing cohort. The section on "Study design" does not explain the study design of this study; this is only an explanation of the JECS. The study design of 'this study' should be clarified. Additionally, please clearly describe the definition of case and control.

Author response: We apologize for this confusion. This study is a cross-sectional study because it used data obtained at a single time point, divided participants into two groups based on exposure, and compared the outcomes in both groups. We used the JECS dataset and investigated the association between cesarean delivery and the development of infectious diseases in infants. We performed multiple logistic regression analyses to determine the risk of infectious diseases associated with cesarean delivery by eliminating confounding factors and calculating odds ratios. We have revised the text of the manuscript as follows:

Page 3, lines 30–32

This study is a cross-sectional study. We used data from the Japan Environment and Children’s Study, which is a prospective, nationwide, government-funded birth cohort study.

Page 5, lines 75–78

This study is a cross-sectional study. We used data from the JECS and investigated the association between cesarean delivery and the development of infectious diseases in infants. The detailed design of the JECS, a prospective nationwide government-funded birth cohort study, has been previously described [20].

Page 7, lines 113–117

Information about infectious diseases was obtained from the C1Y response to “Has your child ever been diagnosed by a doctor with the following infections?” The following diseases were targeted: central nervous system infection (CNSI), OM, upper respiratory tract infection (URTI), LRTI, gastrointestinal infection (GI), and urinary tract infection (UTI).

Page 7, lines 120–122

The mode of delivery was divided into cesarean sections and vaginal deliveries based on medical record transcripts in the Dr0m data. Participants were assigned to the cesarean or vaginal delivery group.

2. Sample size calculation

How did the authors estimate the sample size? Did the authors try to do matching cases and controls to increase the power?

In a nested case-control study, controls are selected by incidence density sampling or sampling from baseline subcohort. Either way, random sampling for controls is used.

It seems that the authors used all cases and all controls available in the main analysis, but how will the result be affected if they used adequate sample size calculation?

Author response: Thank you for your valuable comments. This study is a cross-sectional study. We apologize for this confusion. The ideal sample size has an event-predictor ratio of 10:1 or greater. For each explanatory variable in a multivariate analysis, there should be at least 10 events. This was not a problem in this study because of its large sample size. We have added the following text to the manuscript:

Page 20, lines 199–201

Moreover, the ideal sample size has an event-predictor ratio of 10:1 or greater. Therefore, this was not a problem in this study because of its large sample size.

---

## [Decision Letter · Decision Letter 2]

2 Feb 2024

Association of cesarean section and infectious outcomes among infants at 1 year of age: Logistic regression analysis using data of 104,065 records from the Japan Environment and Children’s Study

PONE-D-23-20377R2

Dear Dr. Maeda,

We’re pleased to inform you that your manuscript has been judged scientifically suitable for publication and will be formally accepted for publication once it meets all outstanding technical requirements.

Kind regards,

Kazumichi Fujioka

Academic Editor

PLOS ONE

Additional Editor Comments (optional):

Congratulations!

Reviewers' comments:

Reviewer's Responses to Questions

**Comments to the Author**

1. If the authors have adequately addressed your comments raised in a previous round of review and you feel that this manuscript is now acceptable for publication, you may indicate that here to bypass the “Comments to the Author” section, enter your conflict of interest statement in the “Confidential to Editor” section, and submit your "Accept" recommendation.

Reviewer #1: All comments have been addressed

2. Is the manuscript technically sound, and do the data support the conclusions?

Reviewer #1: Yes

3. Has the statistical analysis been performed appropriately and rigorously? 

Reviewer #1: Yes

4. Have the authors made all data underlying the findings in their manuscript fully available?

Reviewer #1: Yes

5. Is the manuscript presented in an intelligible fashion and written in standard English?

Reviewer #1: Yes

6. Review Comments to the Author

Reviewer #1: (No Response)

7. PLOS authors have the option to publish the peer review history of their article (what does this mean?). If published, this will include your full peer review and any attached files.

Reviewer #1: No

---

## [Editor Report · Acceptance letter]

12 Feb 2024

PONE-D-23-20377R2 

PLOS ONE

Dear Dr. Maeda, 

I'm pleased to inform you that your manuscript has been deemed suitable for publication in PLOS ONE. Congratulations! Your manuscript is now being handed over to our production team.

Kind regards, 

on behalf of

Dr. Kazumichi Fujioka 

Academic Editor

PLOS ONE